# SOFT TOKEN MATCHING FOR INTERPRETABLE LOW-RESOURCE CLASSIFICATION

## ABSTRACT

We propose a model to tackle classification tasks in the presence of very little training data. To this aim, we introduce a novel matching mechanism to focus on elements of the input, by using vectors that represent semantically meaningful concepts for the task at hand. By leveraging highlighted portions of the training data, a simple, yet effective, error boosting technique guides the learning process. In practice, it increases the error associated to relevant parts of the input by a given factor. Results on text classification tasks confirm the benefits of the proposed approach in both balanced and unbalanced cases, thus being of practical use when labeling new examples is expensive. In addition, the model is interpretable, as it allows for human inspection of the learned weights.

## 1 INTRODUCTION

Gathering and labeling data is a task that can be expensive in terms of time, human effort and resources. When we cannot rely on already available datasets, training a model with acceptable performance on few data points annotated by few annotators, becomes critical in many practical applications. This is, indeed, especially important when the data is naturally imbalanced and the demands of gathering samples of the minority class are high. One important domain in which these issues arise is text classification, for example hate-speech (Waseem & Hovy, 2016), web spam (Castillo et al., 2007) and abuse detection (Mishra et al., 2018).

One effective approach to overcome the lack of training data is that of Zaidan et al. (2007), which consists of augmenting the few data available with *rationales*, i.e., highlighted portions of the input. Rationales are usually coupled with feature-engineering to be effective in low resource scenarios.
An alternative way to deal with data sparsity, especially in text classification tasks, is to use pre-trained language models (LMs) that are fine-tuned on a target domain. While this approach has been tested on hundreds of training points (Devlin et al., 2018; Howard & Ruder, 2018), it is not clear how it behaves in an even scarcer setting, as the vast parameter space of an LM might pose a problem. Moreover, fine-tuning a model may require a considerable amount of computing power, therefore restricting its applicability.
On the other hand, some embedding-based models represent the input as a weighted average of words (Kalchbrenner et al., 2014; Sheikh et al., 2016), where the weight is given by a parameter called "reference vector". However, these models cannot easily incorporate multiple reference vectors, and they are not interpretable since classification works on unreadable embedding features.

In this paper, we propose a novel and efficient representation learning model to address the above issues. The underlying idea is to focus on relevant words in the input while being able to generalise to semantically similar concepts; this is something akin to what a human would do in the presence of data scarcity. We therefore introduce two techniques that should coexist to reflect our intuition. First of all, the model to focuses on specific words by computing *soft* matching probabilities between each word and multiple vectors which represent semantic concepts. Secondly, we guide the learning process to learn *important* concepts thanks to an error boosting technique that exploits rationales. Basically, it encourages the model to reduce the overall error by improving the prediction associated with highlighted words.
Additionally, by direct inspection of model weights it is possible to understand what words it focuses on; in short, the model is *interpretable*[1]. Results across a consistent number of baselines and three

---

[1]See (Guidotti et al., 2019) for the notion of interpretability we use throughout the paper.

datasets also indicate a significant improvement in performance. Interestingly, we always outperform fine-tuned models when little training data is available. Our model can also assist users to train a classifier for a very specific task. As an example, consider training an abstract filtering system with rationales provided by the user itself. The model will then learn to filter out papers that are not matching the user's preferences.

The rest of the paper is structured as follows: Section 2 provides an overview of the existing literature, highlighting similar and different ideas; Section 3 formally introduces the problem as well as our model, providing intuition behind our architectural choices; Section 4 details our experiments and shows our findings, with a thorough ablation study that disentangles the contribution of each part of the model and a use case on interpretability; finally, Section 5 summarizes our work.

## 2 RELATED WORKS

There are different ways in which rationales can be used. Some works *generate* rationales, while others *exploit* them to inform the learning process. The method proposed by Lei et al. (2016) tackles text classification by learning a distribution of rationales given the text and a distribution of the target class given the rationales. Interestingly, an additional regularization term is added to the loss to produce rationales that are short and coherent. The model makes use of high-capacity recurrent neural networks (Schuster & Paliwal, 1997), thus it is tested on large amounts of training data to prevent overfitting. This work was later refined by Bastings et al. (2019), who proposed a probabilistic version of a similar architecture, where a latent model is responsible for the generation of *discrete* rationales. The main advantage of predicting discrete rationales is that it is possible to constrain their maximum number per sample, thus effectively controlling sparsity. However, it usually requires a large amount of data points to be effective.

The first to exploit rationales in a low resource scenario were Zaidan et al. (2007) and Zaidan & Eisner (2008), by means of a rationale-constrained SVM (Cortes & Vapnik, 1995) and a probabilistic model. Moreover, the latter is realized as a log-linear classifier with heavy use of *feature-engineering*. On the other hand, when rationales are defined on features rather than on samples, one can use the Generalized Expectation (GE) criteria (Druck et al., 2007; McCallum et al., 2007) to significantly improve the performance of classifiers.

Rationales can also be incorporated in the loss function (Barrett et al., 2018), where the attention module (Vaswani et al., 2017) on top of an LSTM (Hochreiter & Schmidhuber, 1997) is forced to attend relevant tokens in a document. This method was not tested on small datasets, possibly because of the aforementioned issues of high capacity models. A similar approach has been successfully applied by Bao et al. (2018) to the weak supervision problem. However, the model assumes one *source domain*, with supervised labels, to learn an attention generation module that is then applied to the target domain. In contrast, our method can be built on a given embedding space with minimum supervision.

Apart from incorporating prior knowledge in the form of rationales, one can augment neural networks with: first-order logic (Li & Srikumar, 2019; Hu et al., 2016a); a corpora of regular expressions (Luo et al., 2018); or massive linguistic constraints (Hu et al., 2016b). While generally powerful and effective, all these methods require domain-specific expertise to define the additional features and constraints that are then explicitly incorporated into the network. In a different manner, the SoPA architecture of Schwartz et al. (2018) learns to match surface patterns on text through a differentiable version of finite state machines. A weighted combination of these patterns is used to classify a document. Instead, BabbleLabble (BL) (Hancock et al., 2018) is a method for generating weak classifiers from natural language explanations when supervision is scarce. These are then fed to Data Programming (DP) (Ratner et al., 2016), a probabilistic framework, that outputs a final score. On one hand, BL works well because it exploits a domain-specific grammar to parse explanations; on the other hand, this grammar must be carefully designed by domain experts.

Finally, the Neural Bag Of Words (NBOW) model (Kalchbrenner et al., 2014) takes an average of token embeddings and applies a logistic regression to classify a document. Its extension, NBOW2 (Sheikh et al., 2016), computes an importance score for each word by comparing it with a single reference vector that is learned. Despite the underlying idea being similar, we propose a different mechanism to focus on relevant words.

In the following, we describe the architecture whose only requirements are i) an embedding space of the input, and ii) additional rationales, though the latter is not strictly required. As we are shall

discuss, the model has a strong inductive bias, which is effective when trained with very little data. Hereinafter, we refer to our new architecture with the name PARCUS, which stands for Pattern Representations on Continuous Spaces.

## 3 THE PARCUS MODEL

Let us consider a classification task in which a very small labelled dataset $\mathcal{D}_{\mathcal{L}} = \{(x_1, r_1, y_1), \ldots, (x_L, r_L, y_L)\}$ is given, where $x_i$ is an input sample, $r_i$ represents the rationale information and $y_i$ is the discrete target label. For the purpose of this paper, an input is a set of tokens $x_i = \{x_i^1, \ldots, x_i^{T_i}\}$ of arbitrary size $T_i$. In addition, $x_i^j \in \mathbb{R}^d$, where $d$ is the size of an embedding space obtained using a pretrained model. Finally, we assume that each token in the input has been marked as relevant or not by annotators, i.e., $r_i = \{r_i^1, \ldots, r_i^{T_i}\} \in \{0, 1\}^{T_i}$.

**Intuition**  When humans are asked to solve a text classification problem after seeing few examples, they tend to look for very simple patterns across the dataset, such as specific words. Nevertheless, humans are also able to generalise to semantically similar concepts; our goal is to design a model that reflects this ability. For example, assume that the word "excellent" is important for classifying a movie review as positive. If we were to work in the character space, a straightforward solution would be to match specific (sub-)strings in the input, an instance of the so-called *pattern matching* technique. Clearly, pattern matching cannot generalize to words that have similar meaning, e.g., "outstanding". In this work, we transfer the concept of pattern matching into the embedding space, where semantically similar words are assumed to have "close" representations. We achieve this via a mechanism that outputs a probability of *soft* matching between a token and a "reference vector", which is learned together with the classification task to capture discriminative "concepts". Differently from bag-of-word methods of Section 2, this model easily accommodates multiple reference vectors, hence it can focus on many different concepts that are critical for classification.

In order to guide the learning process using the given rationales, it seems sensible to magnify the error for those words that have been marked as relevant by annotators. Notwithstanding the simplicity of the idea, the underlying challenge is to effectively embed human knowledge into the reference vectors, which are responsible for the soft matching technique. In other words, the probabilities of soft matching should be highly correlated with the target class.

Finally, we require that classification should be done in such a way that a user can *explicitly* understand which reference vectors are more important for positive or negative prediction. This last step is obtained by learning a linear combination of the soft matching probabilities.

The next sections describe the proposed model in depth. First, we show how to compute and combine soft token matching probabilities, and then we introduce the error boosting technique that incorporates rationales in the training process. It is worth mentioning that both techniques have been designed to coexist, even though the latter is not strictly necessary to train the model.

### 3.1 SOFT TOKEN MATCHING

We now present the core mechanism that implements soft token matching. Let us define a set of parameters $\mathcal{P} = \{p_1, \ldots, p_N\}, \ p_k \in \mathbb{R}^d$ called *prototypes*, where $N$ is an hyper-parameter of the model. A prototype plays a similar role as the reference vector in Sheikh et al. (2016). Going back to our movie review example, one $p_i \in \mathcal{P}$ should ideally adapt to be *close* to the embedding of the word "excellent".

To learn the $N$ prototypes, we employ the *cosine similarity* metric. Cosine similarity can be seen as a way to measure semantic similarity; its co-domain ranges from $-1$, i.e., opposite in meaning, to 1, i.e., same meaning, with 0 indicating uncorrelation. Ideally, we would like our prototypes to have near-1 similarity with the relevant tokens in the input. To this aim, we further define a *gate activation* function $g : [-1, 1] \to [0, 1]$ that takes the distance between a token $x_i^j$ and a prototype $p_k$ and outputs a probability of soft matching:

$$P(x_i^j \text{ soft-matches } p_k) = g(d(x_i^j, p_k)) = a^{d(x_i^j, p_k)-1} \in [0, 1] \tag{1}$$

where $a$ is an hyper-parameter. In practice, the closer to 1 the similarity is, the greater the output of this gated activation, and $g(v) = 1 \Leftrightarrow v = 1$. By choosing a high value of $a$ we strongly penalize

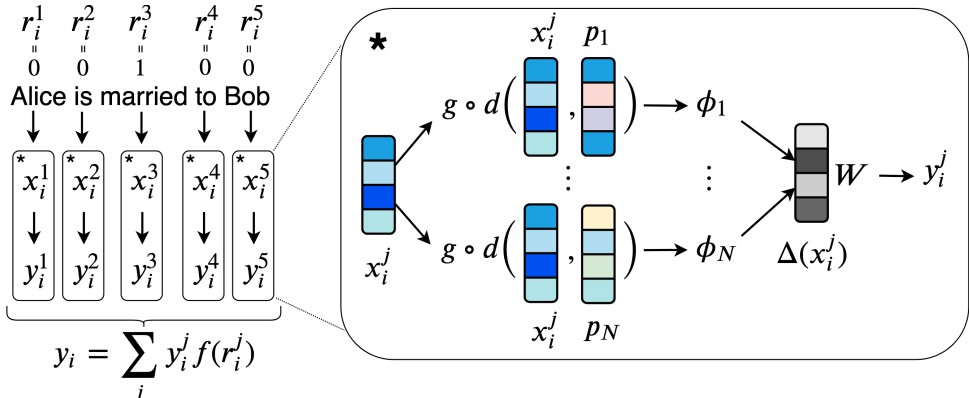

Figure 1: The PARCUS architecture is applied to the $i$-th example of a dataset, i.e., "Alice is married to Bob", with "married" being highlighted. After an initial phase where rationales $r_i$ are extracted and words are mapped to embeddings $x_i$, we extract features by computing the similarity between the token's embedding and the parameters of our model. Then, we combine these features with a linear layer which outputs per-token predictions. At training time only, predictions are multiplied by a boosting factor $f(r_i^j)$. Results are then summed to yield the sentence prediction $y_i$.

tokens that are associated with low similarity scores. For completeness, Section A.1 depicts $g(v)$ for different values of $a$. Such a technique is important because it allows PARCUS to focus on $N$ concepts that are semantically different while fostering interpretability. Notice that this method differs from NBOW2 (Sheikh et al., 2016), as we use prototypes to compute per-token features rather than importance scores.

## 3.2 COMBINING PROTOTYPES

Equation 1 computes the probability of soft matching between a token and a prototype. Likewise, because we have $N$ prototypes, we treat all $N$ probabilities as features associated with that token. We represent these features as $\phi_k(x_i^j) = g(d(x_i^j, p_k)) \; \forall k \in 1, \dots, N$. Now that we have a notion of multiple soft matching probabilities, we can combine them via AND/OR logical functions. An approximation of such functions can be straightforwardly implemented through the pseudo-differentiable version of $min$ and $max$ (Paszke et al., 2017):

$$\phi_{AND}(x_i^j) = min(\{\phi_k(x_i^j) \;\; \forall k\}) \tag{2}$$

$$\phi_{OR}(x_i^j) = max(\{\phi_k(x_i^j) \;\; \forall k\}) \tag{3}$$

In Section A.2 we propose a fully differentiable version of the above equations, though $min$ and $max$ significantly speed up convergence (due to the absence of non-linearities).

## 3.3 INFERENCE

Finally, we need to linearly combine all $F$ features to output a token prediction $y_i^j$. Let us define an auxiliary function (omitting the argument $x_i^j$ to make notation less cluttered):

$$\Delta(x_i^j) = [\phi_1, \dots, \phi_N, \phi_{AND}, \phi_{OR}] \in [0, 1]^{1 \times F} \tag{4}$$

where square brackets denote concatenation. Then, the token prediction is computed as

$$y_i^j = \Delta(x_i^j)\mathbf{W} + b \tag{5}$$

where $\mathbf{W} \in \mathbb{R}^{F \times C}$ is a matrix of parameters (multi-class prediction with $C$ classes') and $b$ is the (optional) bias. It is worth noticing that these features are yet another strong inductive bias, and that the linear model is especially necessary to interpret the model, as detailed in Section 4.3. Figure 1 combines all steps of the proposed architecture for token classification.

Finally, the input prediction is just a sum of the individual $y_i^j$

$$y_i = \sigma(\sum_j^{T_i} y_i^j), \tag{6}$$

where $\sigma$, from now on, represents the *softmax* activation.

## 3.4 Rationale-driven Error Boosting

So far, we have not made use of rationales, which are of fundamental importance to guide the learning process. Intuitively, we would like the prototypes to softly match those tokens that are *relevant for prediction*. It follows that a boosting approach (Freund et al., 1999) is not feasible in this scenario, because we want to weight the importance of tokens rather than of whole samples. Instead, the method we propose is simple and efficient, and it effectively exploits prior information. The idea is to boost the error associated with specific tokens' predictions to encourage the model to focus on them. To be more precise, *at training time only* we modify Equation 6 to take into account the prior information as follows

$$y_i = \sigma(\sum_j^{T_i} y_i^j \cdot f(r_i^j)), \tag{7}$$

where $f : [0, 1] \to \mathbb{R}$ is an arbitrary *exponential* function of our choice that boosts the error, e.g., $f(x) = e^x$; we leave the extension to an adaptive version of $f$ for future works. In terms of learning, $f(r_i^j)$ boosts the gradient of highlighted tokens while leaving unchanged the rest (i.e., if $r_i^j$ is 0, our $f(r_i^j)$ outputs a multiplicative factor of 1).

**Discussion** Regularization of the matrix $\mathbf{W}$ plays an important role for effective learning. We use L1 and L2 regularization terms on $\mathbf{W}$, as done in (Zou & Hastie, 2005), for two main purposes. First, the L1 term enforces sparsity, which allows us to more easily interpret the importance of different features. Secondly, L2 limits the magnitude of the weights, hence avoiding over-compensation of low cosine similarity scores. Consequently, in order to increase one of the soft-matching probability features, the model is encouraged to make changes to the prototypes rather than to the linear weights; in turn, this translates into a particular prototype being "close" to relevant embeddings.

From a mathematical standpoint, we cannot achieve the same result as Equation 6 by means of an additional loss term as done in Lei et al. (2016), because gradients would be summed and not multiplied. Moreover, in our experiments we choose to augment $\Delta(x_i^j)$ with additional information, such as the probability of "opposite" matching: $\phi_{\neg k}(x_i^j) = g(-d(x_i^j, p_k)) \quad \forall k \in 1, \dots, N$. Specifically, when $\phi_{\neg k}(x_i^j) \approx 1$ the token $x_i^j$ and $p_k$ have cosine similarity equal to -1, hence they are opposite in meaning.

**Implementation details** PARCUS can be trained by full-gradient descent in an end-to-end fashion, from the prototypes to the linear weights. We rely on Pytorch (Paszke et al., 2017) to implement our model in few lines of code; this reflects the simplicity and strong inductive bias of our approach, which is necessary in the context we consider. The error boosting technique is applied by the automatic differentiation packages once we implement Equation 7.

We conclude with remarks on the model complexity. The total number of parameters is $\Theta(Nd + FC)$, which is larger than those needed by simpler models such as logistic regression. Usually, a restricted number of parameters serves to counteract overfitting, by limiting the hypotheses space of the model (Vapnik, 1998). However, this work tackles the problem from a different and novel perspective, as we prevent the prototype weights from *freely* changing. Specifically, prototype weights vary in a way that depends on the given embedding space, because they tend to be close to some token representation. If this had not been the case, we could have simply used an MLP in place of $\mathcal{P}$, which does not perform as well as PARCUS in our experiments.

## 4 Experiments

This section reports the experimental setting as well as our experimental findings. We perform an in-depth analysis on our model through ablation studies, in order to clearly separate the contribution

of prototypes from the error boosting technique. Then, we explain what a model can learn by direct inspection of its parameters. All code to reproduce experiments is publicly available[2].

## 4.1 EXPERIMENTAL SETTING

**Datasets** We empirically validate our method on three different datasets. First, MovieReview (Zaidan et al., 2007) contains balanced positive and negative movie reviews with rationales. Secondly, we use the highly imbalanced Spouse dataset from Hancock et al. (2018), where the task is to tell whether two entities in a given piece of news are married or not. This is a much harder task than standard classification, as the same document can appear multiple times with different given entities and background context greatly varies. Finally, we use the Hatespeech Twitter dataset of Waseem & Hovy (2016), which contains short and noisy tweets that can belong to the *hate-speech* or *neutral* class. Datasets' statistics are reported in the appendix for completeness. We manually provide rationales for 60 randomly chosen positive samples of both Spouse and Hatespeech (this process required approximately 1 hour).

**Setup** The experimental evaluation was carried out by measuring performances on the given test set while varying the number of data points used for training. We used balanced train and validation splits for all models, and the validation set is taken as big as the training one to simulate a real scenario. As for Spouse, we used the given validation set to fairly compare with the results of Hancock et al. (2018). We chose the pre-trained *base* version of BERT (Devlin et al., 2018) to provide the embedding space to our method and to other neural baselines as well.
We repeated each experiments 10 times with different random train/validation splits; however, different models have been trained and validated on the *same* data splits, and we report the hyperparameters table in the appendix. Moreover, to avoid bad initializations of the final re-training (for the selected configuration), we average test performances over 3 runs. The optimized measure is Accuracy for MovieReview and F1-score for Spouse and Hatespeech, as the former is perfectly balanced. We optimize the Cross Entropy loss using the Adam optimizer (Kingma & Ba, 2015) for all the baselines we implement.

**Methods** To have a fair evaluation with respect to the same embedding space, we train a linear model (Linear) and a single-layer MLP that work on token embeddings (MLP), as well as NBOW and NBOW2. Importantly, we also finetune BERT on Spouse and MovieReview. For Spouse, we propose a regular expression that associates specific sub-strings (*"wife"*, *"husb"*, *"marr"* and *"knot"*) to the positive class; ideally, our model should be able to focus on such words while also generalizing. Traditional Supervision (TS) is a logistic regression trained on n-gram features, whereas BL-DM stands for the BabbleLabble pipeline tested on 30 random explanations; results for TS and BL-DM are taken from Hancock et al. (2018). BL-DM can explicitly exploit the relational information of the Spouse dataset, hence it is a strong baseline.
Moreover, we report results of an SVM (Zaidan et al., 2007) and a log-linear model on language features (Zaidan & Eisner, 2008), both of which are specifically designed to exploit additional rationales. On Hatespeech, we compare against a Logistic Regression model based on character n-grams (LR-ngrams), as it was shown to reach state of the art performance (Park & Fung, 2017). Finally, we perform a number of ablation studies to isolate the contribution of different techniques: i) an MLP with the error boosting technique; ii) our method without highlights; ii) our method with no logical features; iii) our method with $\phi_k$ features only.

## 4.2 RESULTS & DISCUSSION

Table 1 presents our empirical results for all three datasets, Results confirm that the choice of a strong inductive bias indeed benefits performances in a very low data regime. On Spouse, our model strongly outperforms other neural baselines and reaches the *manually tuned* regular expression with just 60 data points (only 30 of them are positive). Moreover, TS needs $\approx$*50x* more data to achieve similar performance, whereas 10 datapoints are sufficient to do better than almost all baselines with a training size of 300, a >*30x* improvement which does not depend on the chosen embedding space. We also found that TS performs much worse than our linear baseline (hence the need for a fair comparison on the embedding space). With 300 datapoints, our model without highlights has an average

---

[2]Link to the code to reproduce the experiments is omitted at review time

Table 1: Results for all three datasets. Standard deviation is shown in brackets. On Spouse and Hatespeech we report the F1 score, whereas we use accuracy to compare models on MovieReview.

SPOUSE

| MODEL/TRAIN SIZE | 10 | 30 | 60 | 150 | 300 | 3K | 10K | |
|---|---|---|---|---|---|---|---|---|
| TUNED REGEXP | - | - | - | - | - | - | - | 40.48 |
| LINEAR | 18.24 (1.3) | 20.56 (1.4) | 22.46 (1.4) | 26.08 (1.1) | 26.12 (1.2) | - | - | - |
| MLP | 17.91 (2.4) | 20.24 (3.1) | 18.34 (0.6) | 23.26 (1.2) | 24.08 (1.3) | - | - | - |
| NBOW | 21.00 (2.3) | 21.84 (1.7) | 24.03 (1.0) | 27.39 (2.0) | 28.15 (1.8) | - | - | - |
| NBOW2 | 19.51 (2.6) | 22.27 (1.9) | 25.87 (1.4) | 29.62 (1.5) | 31.66 (2.1) | - | - | - |
| BERT+FINETUNING | 16.91 (2.6) | 20.19 (2.1) | 23.41 (1.2) | 32.11 (2.0) | 35.45 (3.2) | - | - | - |
| (Abl.) MLP W. H. | 16.71 (1.4) | 20.81 (2.7) | 20.92 (1.6) | 22.72 (1.8) | 23.12 (2.0) | - | - | - |
| (Abl.) PARCUS WO H. | 27.00 (2.2) | 31.60 (2.5) | 34.24 (2.3) | 41.79 (2.1) | **44.03** (1.2) | - | - | - |
| (Abl.) PARCUS-$\phi_k$ | 32.43 (4.5) | 34.36 (4.2) | 37.76 (2.7) | 42.70 (1.0) | 41.41 (2.4) | - | - | - |
| (Abl.) PARCUS-NO-LOGIC | 32.70 (3.4) | 34.52 (3.9) | 36.76 (2.6) | 42.66 (1.6) | 41.96 (1.9) | - | - | - |
| TS | - | 15.5 | 15.9 | 16.4 | 17.2 | 41.8 | 55.0 | |
| BL-DM (30 EXPL.) | - | - | - | - | - | - | - | **46.5** |
| PARCUS | **34.03** (4.5) | **36.64** (4.3) | **40.29** (2.5) | **43.67** (1.7) | 42.85 (1.6) | - | - | - |

MOVIEREVIEW

| | 10 | 20 | 50 | 100 | 200 |
|---|---|---|---|---|---|
| LINEAR | 60.37 (3.4) | 64.03 (3.5) | 70.17 (2.0) | 77.2 (2.6) | 80.28 (3.1) |
| MLP | 59.12 (4.1) | 62.57 (4.2) | 69.65 (2.4) | 73.25 (3.8) | 80.01 (3.0) |
| NBOW | 62.56 (4.6) | 65.7 (4.8) | 73.88 (1.6) | 77.99 (2.0) | 81.22 (3.6) |
| NBOW2 | 61.45 (4.5) | 64.30 (4.9) | 72.88 (1.4) | 78.87 (4.4) | 83.63 (1.8) |
| BERT+FINETUNING | 53.52 (2.0) | 54.83 (4.9) | 59.72 (4.5) | 67.73 (4.3) | 79.20 (2.5) |
| (Abl.) MLP W. H. | 61.45 (4.6) | 63.05 (5.8) | 68.87 (7.0) | 72.38 (8.5) | 74.63 (5.8) |
| (Abl.) PARCUS WO H. | 61.2 (4.3) | 64.85 (5.0) | 74.25 (2.4) | 78.55 (2.3) | **84.55** (2.8) |
| (Abl.) PARCUS-$\phi_k$ | 66.05 (5.7) | 68.4 (3.5) | **77.8** (2.0) | 80.7 (3.0) | 83.40 (2.4) |
| (Abl.) PARCUS-NO-LOGIC | 66.90 (5.9) | 67.9 (3.5) | 75.5 (4.0) | **80.95** (2.4) | 83.65 (2.7) |
| SVM + RATIONALES | - | 65.4 | - | 75 | 83.2 |
| LOG-LINEAR + RATIONALES | - | 65.8 | - | 76 | 83.75 |
| PARCUS | **67.2** (5.5) | **70.05** (5.6) | 76.55 (2.4) | 79.95 (2.6) | 83.75 (2.8) |

HATESPEECH

| | 10 | 20 | 50 | 100 | 200 |
|---|---|---|---|---|---|
| LINEAR | 37.81 (5.8) | 40.8 (6.2) | 47.35 (3.4) | 51.49 (3.3) | 51.86 (2.0) |
| MLP | 42.09 (8.5) | 42.94 (6.3) | 50.77 (4.2) | 51.03 (5.7) | 54.62 (4.8) |
| NBOW | 41.90 (6.1) | 42.50 (5.7) | 49.07 (3.0) | 52.52 (1.5) | 56.17 (2.5) |
| NBOW2 | 40.20 (6.9) | 41.63 (5.5) | 47.96 (3.2) | 51.52 (2.5) | 53.48 (2.5) |
| LR-NGRAMS | 41.48 (9.1) | **48.00** (7.7) | 49.86 (7.2) | **56.83** (5.5) | **62.75** (1.5) |
| PARCUS | **43.06** (12.7) | 45.25 (9.8) | 48.32 (6.6) | 55.09 (3.0) | 56.39 (4.7) |
| PARCUS (Fasttext) | 40.97 (11.1) | 46.43 (5.3) | **52.46** (4.4) | 54.04 (5.0) | 59.07 (2.3) |

F1 score very close to that of BL-DM, which relies on a domain-specific grammar and parser. In addition, we note that the reported result (46.5) is not averaged over multiple runs; as a matter of fact, one of our random splits achieves a test score of 46.3, indicating the need for robust evaluation when it comes to experiments on few examples. Overall, we found that the proposed approach can be really helpful when data is greatly imbalanced, and outperforms models like BERT that are deemed quite performing when fine-tuned on relatively small datasets (Devlin et al., 2018; Howard & Ruder, 2018). Similar arguments apply to MovieReview, where our model strongly improves over the baselines. Interestingly, our simple representation learning approach is able to beat the state of the art by a large margin when few data points are available. Here, NBOW and NBOW2 models proved to be strong baselines, as the mean representation of a document seems to work well. Generally speaking, the gap between performances is more evident as training size is very scarce, even when compared to other baselines that use rationales.

In light of these results, we performed ablation studies on both datasets to understand if the improvements are only due to prototypes, rationales or both. Overall, we observe that the strong inductive

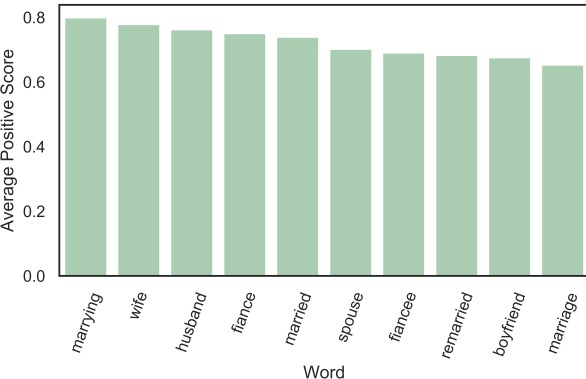

Figure 2: Top-10 most relevant tokens for positive prediction, averaged on *unseen* data.

bias represented by prototypes provides a consistent improvement with respect to the other models, which is especially evident on the Spouse dataset. Interestingly, the MLP does not benefit from error boosting, which might be explained by the fact that its larger hypotheses space, i.e., unconstrained weights, makes it difficult to diminish the contributions of non-relevant tokens. Because rationales guide the learning process, their are more important in the extremely low resource scenario, but their effect slowly fades as the training size increases; contrarily to our expectations, PARCUS performed even better on larger amounts of training points without rationales.

On Hatespeech PARCUS still performs fairly well on average, but with 200 traning examples it cannot keep up with the logistic regressor of Park & Fung (2017). We found two reasons for such behavior: (i) BERT's tokenizer is unable to accurately splitting tweets, due to their noisy nature, and (ii) character n-grams are strong discriminative features for this task (Nobata et al., 2016), and BERT does not use them. To simultaneously solve both issues we switched to Fasttext embeddings (Bojanowski et al., 2017) which are also trained on character n-grams and do not need an additional tokenizer. Surprisingly, we observed a significant improvement on 200 data points, which reduces the gap in performance with the n-grams based model. Therefore, we conclude that choosing the "right" embedding space can make the difference, which does not necessarily mean using one of the latest and most powerful language models available.

## 4.3 PROVIDING EXPLANATIONS

In this Section we show that PARCUS is interpretable. To this aim, we train a model using $N{=}3$ prototypes on 60 examples taken from the Spouse dataset. Then we run the model on *unseen* data and we inspect the outputs associated with each token. We then rank them to see what are the most important ones, and we observe that the tokens with highest rank correspond to semantic concepts that are relevant for the task. Indeed, the model learned to focus on words related to marriage, as well as syntactic variations associated with similar semantics. Moreover, some of the words were not given as part of rationales in the training set. The next step is to show that rationales have effectively been incorporated in the prototypes, and how the features of Eq. 4 have been weighted. We started by inspecting the magnitude of the linear weights $\mathbf{W} \in R^{8 \times 2}$; specifically, if the $i$-th feature is discriminative for a class $c$, then the $i$-th row of $\mathbf{W}$ will have the $c$-th element clearly larger than the others. In our example, we found that $\phi_1$ was important for positive predictions, whereas the other features did not contribute to a particular class. Therefore, we performed top-10 cosine similarity ranking between tokens and the prototype $p_1$. From the most similar to the least one, we obtained: *husband*; *marriage*; *marrying*; *wife*; *married*; *marry*; *fiance*; *wedding*; *fiancee*; and *girlfriend*. Interestingly, PARCUS has automatically learned to match concepts similar to those provided in natural language form by BabbleLabble Hancock et al. (2018).

## 5 CONCLUSIONS

We presented a new methodology to perform classification in the low data regime. We coupled soft token matching with error boosting to focus on concepts that are important for the task at hand. The model is able to outperform other competitors, including fine-tuned complex models. Moreover, we showed with a practical example how humans can interpret the predictions in terms of concepts matching. In conclusion, our model proved to be a useful in tasks where gathering data is challenging.

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

## A  APPENDIX

### A.1  GATING ACTIVATION FUNCTION

The gating activation function of Eq. 1 controls how similar an embedding and a prototype have to be in order to significantly contribute as a feature. Figure 3 shows different curves for different values of $a$. A bigger $a$ squashes low cosine similarities more, as is the case for $a = 100$, therefore acting as a stricter filter. For completeness, Figure 5 shows the effect of a g($\cdot$) in $R^2$, using $a = 100$.

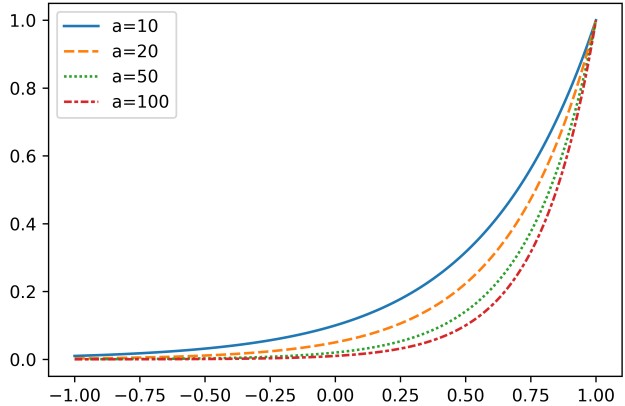

Figure 3: Plot of the gating function $g(v) = a^{v-1}$ for different values of $a$. In practice, the argument $v$ will be the cosine similarity between 2 vectors, hence $g : [-1, 1] \to [0, 1]$

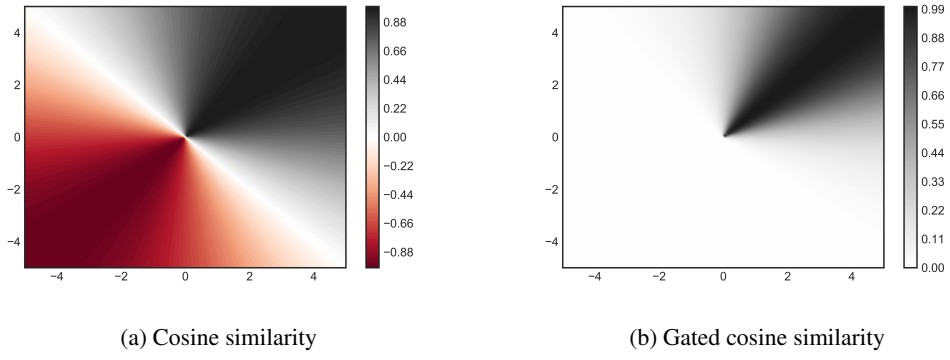

(a) Cosine similarity                    (b) Gated cosine similarity

Figure 4: From left to right: (a) Cosine similarity with respect to the [1,1] vector $v$ and (b) its gated activation for $a = 100$. The gate function acts as a filter on cosine similarity, thus promoting those vectors "close enough" to $v$ while penalizing the others.

## A.2 DIFFERENTIABLE IMPLEMENTATION OF AND AND OR

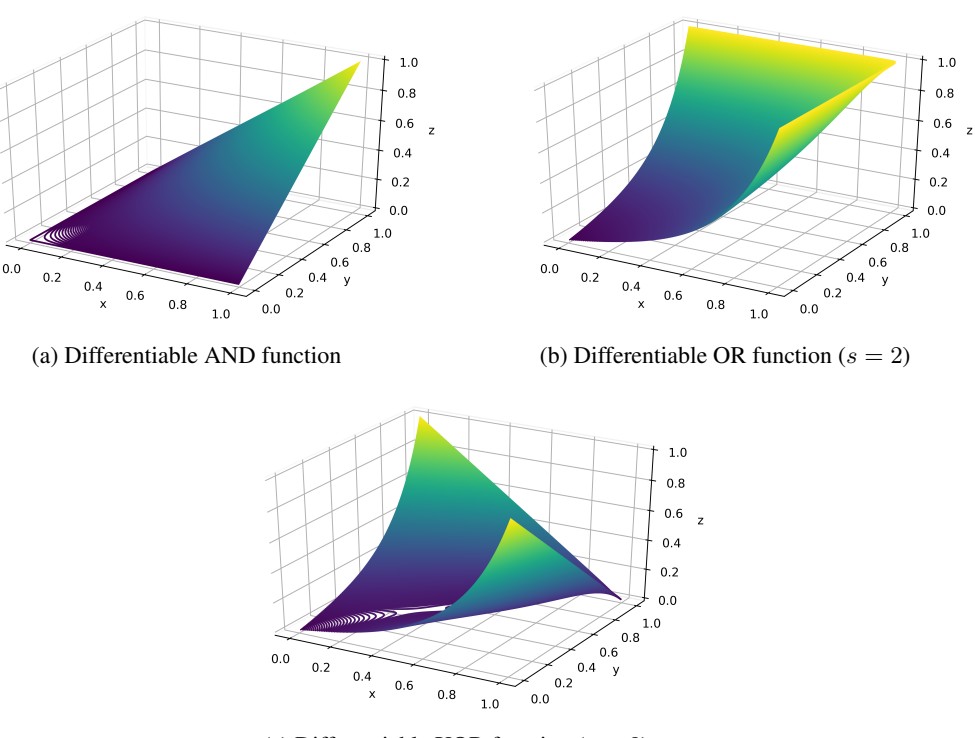

(a) Differentiable AND function

(b) Differentiable OR function ($s = 2$)

(c) Differentiable XOR function ($s = 2$)

Figure 5: From left to right: Cosine similarity with respect to the [1,1] vector and its gated activation.

Given the set $\{\phi_1, , \phi_N\} \in [0, 1]^N$, interpreted as $N$ *independent* "soft matching probabilities", we would like to compute their joint probability as well as the probability of a single matching out of all possible ones. In our experiments, we found a pseudo-differentiable implementation of $min$ and $max$ (Eq. 2 and Eq. 3) to speed up convergence. Alternatively, we tried a fully differentiable version of those two functions as well as the probability of mutually exclusive events, i.e., XOR. We present the equations for $N = 2$, although this can be easily generalized to arbitrary $N$:

$$\phi_{AND}^{Diff}(\phi_1, \phi_2) = \qquad\qquad\qquad\qquad\qquad\qquad\qquad\qquad\qquad \phi_1 * \phi_2$$

$$\phi_{OR}^{Diff}(\phi_1, \phi_2, s) = \quad (\phi_1 - 2)^{-2s} - 2^{-2s}(1 - \phi_1) + (\phi_2 - 2)^{-2s} - 2^{-2s}(1 - \phi_2) - \phi_{AND}^{Diff}(\phi_1, \phi_2)$$

$$\phi_{XOR}^{Diff}(\phi_1, \phi_2) = \qquad\qquad\qquad\qquad\qquad\qquad \phi_{OR}^{Diff}(\phi_1, \phi_2) - \phi_{AND}^{Diff}(\phi_1, \phi_2)$$

where $s$ controls how squashed is the curve is. Figure 5 depicts all three curves for $s = 2$.

## A.3 DATASETS' STATISTICS

In Table 2 we report the statistics of the datasets we used. When preprocessing the data with BERT, we used a maximum sentence length of 128 for token-based methods, and 512 when finetuning BERT.

Table 2: Datasets' statistics.

|  | Train Size | Validation Size | Test Size | % Pos. | No. Rationales |
|---|---|---|---|---|---|
| Spouse | 22195 | 2796 | 2697 | 8% | 60 (Positive) |
| MovieReview | 1800 | - | 200 | 50% | 1800 |
| Hatespeech | 3430 | - | 3430 | 14.7% | 60 (Positive) |

As mentioned in Section 4, each sample of the Spouse dataset contains a pair of entities as well as the sentence. Therefore, the MLP baseline, ablation studies and our method will make use of an input-mask (at test time only), which reflects the fact that we are not interested in those sentences where both our entities of interest do not appear. It is worth noticing that such relational information should be naturally exploited by methods like BabbleLabble, which rely on domain-specific grammars. While this method should help to improve precision, in practice it did not significantly affect performances.

## A.4 HYPER-PARAMETERS

Hyper-parameters are used to perform model selection, which returns the best configuration for a given validation split. We use such configuration to train a model on the whole training set and then evaluate its generalization performances on the unseen test set. When Data Programming is used, we simply combine model selection of our model with that of data programming. Since the experiment is repeated 10 times to avoid lucky/unlucky data splits, model selection is performed 10 times as well. Model assessment, i.e., the measure of performance of our family of models, is evaluated by averaging the 10 different performances on the test set. When fine-tuning BERT, we mainly follow the guidelines reported in Devlin et al. (2018).

Table 3: Hyper-parameters tried for model selection.

|  | LINEAR | MLP/NBOW(2) | BERT+FINETUNE | DP | OURS |
|---|---|---|---|---|---|
| LEARNING RATE | $\{0.01, 0.001, 0.0001\}$ | $\{0.001, 0.0001\}$ | $\{0.00002, 0.00003, 0.00005\}$ | $range(0.001, 0.0001)$ | $\{0.01\}$ |
| L1 | - | - | - | | - $\{0.01, 0.001\}$ |
| L2 | $\{0.1, 0.01, 0.0001\}$ | $\{0.01, 0.0001\}$ | - | - | $\{0.001, 0.0001\}$ |
| EPOCHS | $\{50, 100, 150\}$ | $\{100, 500\}$ | $\{2, 4, 10\}$ | $\{100, 500\}$ | $\{500\}$ |
| HIDDEN UNITS | - | $\{8, 16, 32\}$ | - | - | - |
| BATCH SIZE | 32 | 32 | 8 | - | 32 |
| $N$ | - | - | - | - | $\{5, 10\}$ |
| $f(r)$ | - | - | - | - | $\{e^r, 5^r, 10^r\}$ |
| $a$ | - | - | - | - | $\{10, 100\}$ |
| MAX SEARCH | - | - | - | 10 | - |

### A.5 Gradient boosting effect on backpropagation

For a loss $\ell$ defined on top of Equation 6 and a true label $\hat{y}$, backpropagation is computed as (abstracting from the sample index $i$ to simplify notation):

$$\frac{\partial \ell(y, \hat{y})}{\partial \boldsymbol{\theta}} = \frac{\partial \ell(y, \hat{y})}{\partial y} \cdot \frac{\partial y}{\partial \boldsymbol{\theta}} = \frac{\partial \ell(y, \hat{y})}{\partial y} \cdot \frac{\partial \sigma(\sum_j^T y^j f(r^j))}{\partial(\sum_j^T y^j f(r^j))} \cdot \frac{\partial \sum_j^T y^j f(r^j)}{\partial \boldsymbol{\theta}}$$

$$\frac{\partial \sum_j^T y^j f(r^j)}{\partial \boldsymbol{\theta}} = \sum_j^T \left( \frac{\partial(\Delta(x^j)\mathbf{W} + b)}{\partial \boldsymbol{\theta}} \cdot f(r^j) \right), \tag{8}$$

where $\boldsymbol{\theta}$ are the parameters of the model.

### A.6 Robustness to Noise

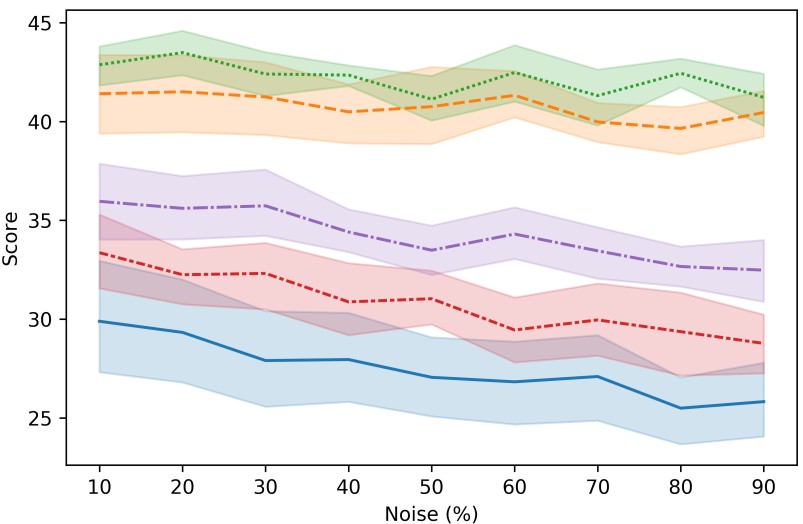

Figure 6: Robustness to different degrees of rationale noise on the Spouse dataset. From bottom to top, the curves refer to different training sizes: 10, 30, 60, 150, 300. We fix $f(r) = e^r$

We investigated how injecting random rationales (i.e., ones) in prior information affects the final performances. Clearly, having a 100% noise corresponds to not having rationales at all, therefore we run a simple experiment (with fixed hyper-parameters) 10 times on Spouse, with the goal to study robustness to noise. Figure 6 shows the result, where different curves stand for different training sizes; as one would expect, for few data points (blue and red curves) noise has bigger influence whereas for 150 (orange curve) and 300 (green curve) data points the effect is negligible. This confirms that there is a trade-off where the impact of rationales has little effect on the final performances.

## A.7 Visualization of the most important tokens

The importance of tokens in a sentence can be inspected by looking at the prototypes $p_i$, the weight matrix $\mathbf{W}$ and the tokens predictions $y_i^j$. In Figure 7 we show the some of the tokens which are responsible for positive prediction on Spouse *unseen* data, which is consistent with the results of Section 4.3.

alex perkins , who is currently in a relationship with presenter anna richardson , was in a relationship with a man for six years after meeting him in school during a theatre production alex told the telegraph she had a long - term relationship with a man who she only calls ' chris , who she met while they both starred in a production of bi ##ze ##t ' s carmen in school .

fast moving couple : more ##na ba ##cca ##rin ' s estranged husband austin chris - pictured back in april 2014 - has made some bold accusations in their bitter custody feud over their one - year - old son regarding her gotham co - star alex . at the centre : her gotham co - star alex , pictured in new york thursday night , is mentioned in court do ##umen ##ts by chris as he claims that he arrived home to gather his belongings and found the 37 - year - old actor ' had just taken a shower and he was playing with my son . ' .

alex and maxi ##ne av ##is e ##wart divorced in 1976 six years after he allegedly had an affair with chris , the second wife of legendary late - night talk show host johnny chris .

alex wondered if the kidnapping of the wife of the managing director of the sun , chris , in lagos last week was also the fault of mimi ##ko .

alex , who is chris ' s second child with husband rand ##e alex and was signed to im ##g models in july , wore a simple quarter - sleeve dress as she sat front row at the cat ##walk show .

alex , the sister of mani ##sha alex who has been found guilty of the murder of pu ##r ##vi joshi , her ex - boyfriend chris new fiance , and is awaiting sentencing .

un ##mo ##ved , u . s . district judge alex ordered bog ##oma ##lova , 52 , taken into custody immediately , and her husband , chris , 62 , returned to prison , where he has been since his arrest .

the night before , kylie and her boyfriend alex attended alexander wang ' s 10 year anniversary show , featuring a star - stud ##ded audience with the likes of chris and lady gaga .

alex and e ##wart ' s daughter victoria married chris son michael in 1981 .

alex is rarely seen with his famous wife and she often jokes about the lengths he goes to to avoid the spotlight , but dolly ' s husband has agreed to be part of her anniversary party in may . alex , 73 , who owns and operates a road repair company in tennessee , will hit the stage with chris , 69 , when she performs at their willow lake plantation home .

close : nicole adopted alex and connor with ex - husband chris in 1992 and 1995 respectively prior to their separation and subsequent divorce in 2001 ( pictured together in 2004 ) . she added : '

pierce is also father to sons alex and christopher from his first marriage to chris , while their daughter charlotte tragic ##ally died in 2013 of o ##var ##ian cancer , the same disease that killed her mother , aged just 42 .

alex couple : nicole was married to chris for 11 - years .

alex ' s wife , chris alex , and their two kids flew to boise to be with him . '

Figure 7: Visualization of the most important tokens for positive prediction. Sentences were extracted from the Spouse dataset. The entities we are interested in are always called "alex" and "chris".

