# OpenReview forum: "Soft Token Matching for Interpretable Low-Resource Classification"
_ICLR.cc/2020/Conference — Reject_

### Official Review · AnonReviewer3 · 2019-10-23
**Official Blind Review #3**

**Rating:** 3

**Review:**

After responses:

I read the authors response and decided to stick to my original score mostly because:

1 - I understand that interpretability is hard to define. I also agree with the authors response. However, this is still not reflected in the paper in any way. I expect a discussion on what is the relevant definition used in the paper and how does it fit to that definition. Currently, it is very confusing to the reader.

2 - I understand the authors' response that few-shot learning is a different empirical setting. However, authors also agree that settings are some-what relevant. I really do not see any gain by NOT discussing the few-shot learning literature. At the end, a reader is interested in this work if they have limited data. Moreover, other ways to address limited data issue should be discussed.

-----
The manuscript is proposing a few-shot classification setting in which training set includes only few examples. The main contribution is using prototype embeddings and representing each word as cosine distances to these prototype embeddings. Moreover, the final classification is weighted summation of the per-token decisions followed by a soft-max. Per-token classifiers are obtained with an MLP using the cosine distances as features. When the relevance labels are available, they are used in training to boost gradients.

PRO(s)
The proposed method is interesting and addressing an important problem. There are many few-shot scenarios and finding good models for them is impactful.

The results are promising and the proposed method is more interpretable than the existing NLP classifiers. I disagree with the claim that the model is interpretable. However, I appreciate the effort to interpret the model.

CON(s)
The model is not interpretable because 1) it starts with embeddings and they are not interpretable, 2) model is full of non-linearities and decision boundaries are not possible to find. In other words, it is not possible to answer "what would make this model predict some other classifier".

The authors should discuss the existing few-shot learning mechanisms. Especially, "Prototypical Networks for Few-shot Learning" is very relevant. I also think it can be included as a baseline with very minimal modifications.

The writing is not complete. The authors do not even discuss how the prototypes are learned. I am assuming it is done using full gradient-descent over all parameters. However, this is not clearly discussed. Implementation details should be discussed more clearly.

SUMMARY
I believe the manuscript is definitely interesting and has a potential. In the mean time, It is not ready for publication. It needs a through review of few-shot learning. Authors should also discuss can any of the few-shot learning methods be included in the experimental study. If the answer is yes, it should be. If the answer is no, it should be explained clearly.

Although my recommendation is weak-reject, I am happy to bump it up if these issues are addressed.

**Experience Assessment:**

I have read many papers in this area.

**Review Assessment: Checking Correctness Of Derivations And Theory:**

I carefully checked the derivations and theory.

**Review Assessment: Checking Correctness Of Experiments:**

I assessed the sensibility of the experiments.

**Review Assessment: Thoroughness In Paper Reading:**

I read the paper at least twice and used my best judgement in assessing the paper.

---

> ### Author Response · Authors · 2019-11-12
> **Authors' Response to Reviewer #3**
>
> We would like to thank the reviewer for his/her constructive comments. We provide an answer to each point raised below.
>
> **QUESTION**
> The model is not interpretable because 1) it starts with embeddings and they are not interpretable, 2) model is full of non-linearities and decision boundaries are not possible to find. In other words, it is not possible to answer "what would make this model predict some other classifier".
>
> **ANSWER**
> We thank the reviewer for letting us clarify this point. It is common knowledge that it is very tricky to define what it means for a model to be interpretable [1].
> We do agree with the reviewer that we cannot interpret the given embedding space and that it is difficult to derive decision boundaries for the model; however, when using the term “interpretable”, we followed the definition of Interpretability given in [2], i.e., “to which extent the model and/or its predictions are human understandable”.
>
> We argue to have provided a model that is interpretable according to this definition. When a sentence is classified, we are able to inspect the individual contribution of each token to the final classification score, and by design the model is forced to assign high scores to relevant tokens, as shown in Figure 2, page 8. Moreover, we can understand which prototypes/logical features led to such an important individual score. In turn, this allows us to understand which prototypes were most relevant, and we are able to match them with semantic concepts associated with pre-trained embeddings, using the mechanism discussed in Section 4.3.
>
> In summary, we are able to give a human-readable suggestion on what semantic concepts are most important in the sentence and how they combine to give the final answer. This allows us to “bypass” the barrier imposed by the presence of non-linearities in the architecture.
>
> Nonetheless, we kindly ask the reviewer for further suggestions on how he/she thinks we should rephrase this concept in the paper if needed. For the time being, we have added a reference to [2] in our paper to clarify what we mean with the term “interpretable”.
>
>
> **QUESTION**
> The authors should discuss the existing few-shot learning mechanisms. Especially, "Prototypical Networks for Few-shot Learning" is very relevant. I also think it can be included as a baseline with very minimal modifications.
>
> **ANSWER**
> We tend to disagree with the reviewer on this matter. Despite similar, we think our experimental setting significantly differs from the few-shot scenario, in which (as also written in [3] and [4]) a classifier must generalize to new classes not seen in the training set. In our scenario, we do not have unseen classes. Moreover, in order to exploit rationales, it is necessary to see the corresponding class. These are the main reasons why we decided not to include few-shot learning mechanisms in the related works in the first place. We believe these two application scenarios have to be clearly separated from each other.
> However, we thank the reviewer for her/his comment as it might have shed light on a future work in which we modify PARCUS to fit the few-shot or zero-shot setting, and we hope our explanation satisfies the reviewer’s concern.
>
>
> **QUESTION**
> The writing is not complete. The authors do not even discuss how the prototypes are learned. I am assuming it is done using full gradient-descent over all parameters. However, this is not clearly discussed. Implementation details should be discussed more clearly.
>
> **ANSWER**
> We thank the reviewer for highlighting this point. We addressed this point in Section 3.

---

> > ### Author Response · Authors · 2019-11-12
> > **References**
> >
> > [1] Lipton, Zachary C. "The mythos of model interpretability." arXiv preprint arXiv:1606.03490 (2016).
> >
> > [2] Guidotti, Riccardo, et al. "A survey of methods for explaining black box models." ACM computing surveys (CSUR) 51.5 (2019): 93.
> >
> > [3] Snell, Jake, Kevin Swersky, and Richard Zemel. "Prototypical networks for few-shot learning." Advances in Neural Information Processing Systems. 2017.
> >
> > [4] Chen, Wei-Yu, et al. A Closer Look at Few-shot Classification.” International Conference on Learning Representations. 2019.

---

### Official Review · AnonReviewer2 · 2019-10-24
**Official Blind Review #2**

**Rating:** 3

**Review:**

The authors propose PARCUS ("Pattern Representations on Continuous Spaces"), a model which computes a soft-matching probability for all words in an input sequence with so-called prototypes in order to predict a label for the input. Furthermore, for training, PARCUS makes use of rationales. Those are indicators of input importance, and help to boost the loss for relevant tokens.

The main motivation to use PARCUS is that it works better in a low-resource setting than recent state-of-the-art models for the high-resource case. This is due to it having relatively few parameters and to it having a strong inductive bias. However, the fact that models with less parameters perform better than BERT-based models in the low-resource case is not very surprising. Looking at the experiments, the results on HATESPEECH show less differences between models than for SPOUSE or MOVIEREVIEW.

Another selling point of PARCUS is that it's interpretable. While neural networks can also be analyzed in different ways, I agree with the authors that this is nice to have.

Overall, the paper seems solid.

==========

Update: After reading the other reviews and the responses by the authors, I lowered my score from 6 to 3.

**Experience Assessment:**

I have read many papers in this area.

**Review Assessment: Checking Correctness Of Derivations And Theory:**

N/A

**Review Assessment: Checking Correctness Of Experiments:**

I assessed the sensibility of the experiments.

**Review Assessment: Thoroughness In Paper Reading:**

I made a quick assessment of this paper.

---

> ### Author Response · Authors · 2019-11-12
> **Authors' Response to Reviewer #2**
>
> We thank the reviewer for mentioning the benefits of the proposed model. We now clarify some of the reviewer’s doubts:
>
> **QUESTION**
> BERT-based models in the low-resource case is not very surprising
>
> **ANSWER**
> While this result may not look surprising, to the best of our knowledge it was not addressed before for the specific case of BERT. In [1], the authors claim that a large pre-trained model can be very helpful in transfer learning scenarios, and they also suggest how to best fine-tune BERT. We followed their guidelines and included the results to provide convincing evidence that, in this extreme scenario, our model can perform better (even without the use of rationales).
>
> **QUESTION**
> Looking at the experiments, the results on HATESPEECH show less differences between models than for SPOUSE or MOVIEREVIEW.
>
> **ANSWER**
> The reviewer is correct. Apart from the reasons mentioned in the paper, it is possible to observe (by manual inspection) that the tweets of the HATESPEECH dataset are very noisy, short and often similar in meaning. This clearly helps models based on n-gram features, as we have argued in our work.
> SPOUSE and MOVIEREVIEW are different in this sense. SPOUSE, which is where PARCUS performs very well, contains sentences of very different nature and context, which makes it very important to focus on specific concepts (hence the use of prototypes seems appropriate). MOVIEREVIEW, on the other hand, contains very long reviews that need “filtering” to highlight the important concepts. This is another context in which PARCUS can be successfully applied, as training a complex model on few data points that contain “lengthy” sentences can be a hard task to solve.

---

### Official Review · AnonReviewer1 · 2019-10-27
**Official Blind Review #1**

**Rating:** 1

**Review:**

This paper considers the problem of text classification, especially the settings in which the number of labeled sentences is very small. However, authors assume, annotations of rationales behind the label, i.e. highlighting tokens in a sentence which are important in deciding its label. As per my understanding, this is a big limitation. Second, the proposed model makes inference of class labels just based upon occurrence of words in a sentence, rather than making more sophisticated inferences relying upon sub-sequence patterns at least.

The idea proposed in the paper is to learn prototype vectors which have high similarity w.r.t. tokens in sentences, especially the highlighted one. I didn't understand the justification for learning such prototypes in the first place.

This works build upon a workshop paper.

The idea proposed in the paper, even in the specific problem context considered, are incremental. I don't think that this kind of work aligns with the theme of learning representations. This paper may be suitable for publication in an NLP workshop as the baseline model.


**Experience Assessment:**

I have read many papers in this area.

**Review Assessment: Checking Correctness Of Derivations And Theory:**

I carefully checked the derivations and theory.

**Review Assessment: Checking Correctness Of Experiments:**

I carefully checked the experiments.

**Review Assessment: Thoroughness In Paper Reading:**

I read the paper thoroughly.

---

> ### Author Response · Authors · 2019-11-12
> **Authors' Response to Reviewer #1**
>
> We thank the reviewer for writing this review. Although we appreciated the interesting insight that has been expressed in the reviewer’s comment (about the extension of our model to sub-sequence patterns), we do respectfully disagree with many of the points that have been raised. In particular, we found that some of the comments are quite difficult to address given that their nature is not particularly constructive.
>
> Some of the questions posed by the reviewer are difficult to contextualize; in particular, it looks like the reviewer might have misunderstood the assumptions and the advantages of the proposed work, despite the reviewer confidence about this review.
>
> The reviewer has never mentioned that this paper lacks clarity of presentation, but all the answers to his/her questions have already been clearly stated and contextualized in the paper. Moreover, some of the reviewer’s claims about the value of this work are not backed up by evidence or detailed motivations.
>
> As we also state below, we think this work should be judged in terms of how it exploits the additional rationales, rather than criticizing their presence in the first place. Nonetheless, in Section 4.2 we also show (via ablation studies) how our model performs well even without the use of rationales.
>
> In light of this premise and of the following comments (which are meant to further clarify our main research questions) we kindly ask the reviewer to reconsider his/her judgment.

---

> > ### Author Response · Authors · 2019-11-12
> > **Authors' Response to Reviewer #1 - Major Points**
> >
> > **QUESTION**
> > This paper considers the problem of text classification, especially the settings in which the number of labeled sentences is very small. However, authors assume annotations of rationales behind the label, i.e. highlighting tokens in a sentence which are important in deciding its label. As per my understanding, this is a big limitation.
> >
> > **ANSWER**
> > We thank the reviewer for giving us the opportunity to clarify this point. This work follows a whole line of research [1,2,3,4] that clearly contextualizes the use of rationales in a low-resource scenario, where having rationales is not a limitation but an additional tool to more effectively solve the task at hand. This is especially true when the effort of gathering data of a rare class is much higher than the time spent highlighting an example in order to guide the learning process. If the reviewer thinks we should stress this point more in the paper we would be happy to do so.
> >
> > Moreover, as mentioned in Section 4.1, it took just 1 hour per dataset to define all the rationales we used to train the model, which is a very small amount of time when contextualized in a low-resource scenario.
> >
> > To summarize, we want to stress that our goal is the study of how to best exploit this kind of additional information to solve low-resource classification tasks. This is one of our main research questions, and as such it is part and parcel of the ideas explored in this paper.
> >
> > Finally, given that the reviewer clearly states that rationales are a big limitation of the paper, we would like to get his/her opinion on this point to further improve our research.
> >
> >
> > **QUESTION**
> > Second, the proposed model makes inference of class labels just based upon occurrence of words in a sentence, rather than making more sophisticated inferences relying upon sub-sequence patterns at least.
> >
> > **ANSWER**
> > Our model makes inference based on the occurrence of specific semantic concepts in the sentence, rather than focusing on words. Dealing with sub-sequence patterns is not straightforwardly applicable to our architecture without having clearly defined what is a semantic concept associated with sub-sequences. As this is no easy task, we stuck to the current solution, which proved to be very effective on the classification tasks with or without the use of rationales as shown in Table 1, page 7.
> >
> >
> > **QUESTION**
> > The idea proposed in the paper is to learn prototype vectors which have high similarity w.r.t. tokens in sentences, especially the highlighted one. I didn't understand the justification for learning such prototypes in the first place.
> >
> > **ANSWER**
> > Prototypes are needed to implement the idea of soft-token matching in the embedding space, which is the main idea and contribution of this paper. The reasons why we introduced them are i) they allow us to impose a strong inductive bias on the architecture, as discussed in Section 3.4; ii) they can be interpreted as semantic concepts thus allowing for human inspection of the learned weights, as discussed in Section 4.3. Both of these reasons are of practical importance for low-resource classification, as the former point serves to regularise the model whereas the latter helps the human understand whether the model has learned something useful or not.
> >
> >
> > **QUESTION**
> > This works build upon a workshop paper.
> >
> > **ANSWER**
> > This work does not build on a workshop paper. It builds on the foundational ideas proposed by [1]. Can the reviewer please specify the work he/she is referring to?
> >
> > The idea proposed in the paper, even in the specific problem context considered, are incremental. I don't think that this kind of work aligns with the theme of learning representations. This paper may be suitable for publication in an NLP workshop as the baseline model.
> >
> > The reviewer has not given any motivation behind such claims. Could he/she please specify why, in his/her opinion, these ideas are incremental?
> >
> > We strongly disagree with this. We feel our model brings novelty or various reasons: i) the way we use prototypes to construct different matching probabilities; ii) the way we combine these prototypes with logical features; iii) the way we try to interpret the model; iv) the way we exploit rationales. We highlighted the differences between our model and many others in Section 2.
> >
> > Moreover, prototypes are trained to align with semantic concepts, that is representations, that are already given. While this is different from generating representations from scratch, we think the ideas of this paper well align with this venue.

---

> > > ### Author Response · Authors · 2019-11-12
> > > **References**
> > >
> > > [1] Omar Zaidan, Jason Eisner, and Christine Piatko. Using annotator rationales to improve machine learning for text categorization. In Human language technologies 2007: The conference of the North American chapter of the association for computational linguistics; proceedings of the main conference, pp. 260–267, 2007.
> > >
> > > [2] Zaidan, Omar F., and Jason Eisner. "Modeling annotators: A generative approach to learning from annotator rationales." Proceedings of the Conference on Empirical Methods in Natural Language Processing. Association for Computational Linguistics, 2008.
> > >
> > > [3] Druck, Gregory, Gideon Mann, and Andrew McCallum. "Learning from labeled features using generalized expectation criteria." Proceedings of the 31st annual international ACM SIGIR conference on Research and development in information retrieval. ACM, 2008.
> > >
> > > [4] Mann, Gideon S., and Andrew McCallum. "Generalized expectation criteria for semi-supervised learning with weakly labeled data." Journal of machine learning research 11.Feb (2010): 955-984.

---

### Decision · Program_Chairs · 2019-12-19

**Decision:**

Reject

**Comment:**

The authors focus on low-resource text classifications tasks augmented with "rationales". They propose a new technique that improves performance over existing approaches and that allows human inspection of the learned weights.

Although the reviewers did not find any major faults with the paper, they were in consensus that the paper should be rejected at this time. Generally, the reviewers' reservations were in terms of novelty and extent of technical contribution.

Given the large number of submissions this year, I am recommending rejection for this paper.